# ChatCoT: Tool-Augmented Chain-of-Thought Reasoning on Chat-based Large Language Models

**Zhipeng Chen[1,3]**, **Kun Zhou[2,3]**, **Beichen Zhang[1,3]**, **Zheng Gong[1,3]**,
**Wayne Xin Zhao[1,3]†** and **Ji-Rong Wen[1,2,3]**

[1]Gaoling School of Artificial Intelligence, Renmin University of China.
[2]School of Information, Renmin University of China.
[3]Beijing Key Laboratory of Big Data Management and Analysis Methods.
{zhipeng_chen,gongzheng0109,jrwen}@ruc.edu.cn, francis_kun_zhou@163.com
{zhangbeichen724,batmanfly}@gmail.com

## Abstract

Although large language models (LLMs) have achieved excellent performance in a variety of evaluation benchmarks, they still struggle in complex reasoning tasks which require specific knowledge and multi-hop reasoning. To improve the reasoning abilities, we propose **ChatCoT**, a tool-augmented chain-of-thought reasoning framework for chat-based LLMs (*e.g.,* ChatGPT). In ChatCoT, we model the chain-of-thought (CoT) reasoning as multi-turn conversations, to utilize tools in a more natural way through chatting. At each turn, LLMs can either interact with tools or perform the reasoning. Our approach can effectively leverage the multi-turn conversation ability of chat-based LLMs, and integrate the thought chain following and tools manipulation in a unified way. Specially, we initialize the early turns of the conversation by the knowledge about tools, tasks, and reasoning format, and propose an iterative *tool-augmented reasoning* step to perform step-by-step tool-augmented reasoning. The experiment results on two complex reasoning datasets (MATH and HotpotQA) have shown the effectiveness of ChatCoT on complex reasoning tasks, achieving a 7.9% relative improvement over the state-of-the-art baseline. Our code and data are available at: https://github.com/RUCAIBOX/ChatCoT.

## 1 Introduction

Recently, large language models (LLMs) (Zhao et al., 2023a) have shown great potential as general-purpose task solvers in a variety of real-world applications. With excellent few-shot and zero-shot ability, LLMs, such as GPT-4 (OpenAI, 2023) and LLaMA (Touvron et al., 2023), can even outperform full-data supervised-tuned models on many tasks with suitable prompting strategies.

Among these prompting strategies, chain-of-thought (CoT) prompting (Wei et al., 2022; Ko-

jima et al., 2022) has been a prominent approach to eliciting the reasoning abilities of LLMs. It incorporates the intermediate reasoning steps of exemplars into the input prompt, to instruct LLMs to solve a question step by step. Despite the remarkable improvement by CoT prompting, LLMs still have difficulties in solving complex reasoning tasks that involve specific functionalities, such as arithmetic calculation and information retrieval (Lu et al., 2022; Qian et al., 2022). To address this issue, external tools (*e.g.,* calculator, search engine) have been employed to fulfill the basic functionalities (Schick et al., 2023; Paranjape et al., 2023), easing the burden of LLMs. With proper interfaces, LLMs can be guided by prompts to manipulate tools when necessary.

However, as tools are not intrinsically integrated with LLMs, incorporating external tools would have to interrupt the CoT reasoning process of LLMs. Such an issue would become more intractable on complex reasoning tasks that frequently invoke the use of tools. To address it, existing work either relies on LLMs to prearrange the tool use plan for subsequent execution (Zhou et al., 2022; Jiang et al., 2023b), or needs to design formal actions pertaining to specific tasks (Dua et al., 2022; Khattab et al., 2022; Jiang et al., 2023a). Despite the effectiveness, the two types of methods still suffer from potential issues: the former one cannot interact with tools after generating the plan, even seeing obvious mistakes; while the latter one has to frequently switch between reasoning with LLMs and taking actions, hurting the continuity of the CoT reasoning process.

To overcome these disadvantages, we seek a more unified way to integrate CoT reasoning and tool manipulation. As the key idea, we consider tools manipulation by LLMs as the *interaction* between LLMs and tools, in which LLMs send the use requests and tools respond to support specific functions. Further, inspired by the recent progress

---

* Equal contributions.
† Corresponding author.

of ChatGPT-like LLMs (called *chat-based LLMs*), we model the interaction process between LLMs and tools as a multi-turn conversation, and leverage the excellent chatting capacities for manipulating tools by LLMs. At each turn, the LLM can freely interact with tools when in need, otherwise perform the reasoning by itself. The conversation continues until the final answer is derived by LLMs. In this process, as chat-based LLMs can well understand the multi-turn context, they can follow the thought chain in the whole conversation and naturally invoke the tools accordingly, thus keeping the continuity of the reasoning process.

To this end, in this paper, we propose ChatCoT, a tool-augmented chain-of-thought reasoning strategy for chat-based LLMs. As the major merit, ChatCoT can perform the CoT reasoning across multi-turn conversation, and freely interact with tools at immediate steps. Concretely, we first store the useful knowledge at early turns of the conversation, including tools, tasks, and multi-turn reasoning format, to help LLMs utilize task-specific knowledge to perform reasoning or manipulate tools. Then, we iterate a specially designed *tool-augmented reasoning* step in which LLMs interact with tools, to perform step-by-step tool-augmented reasoning, until obtaining the final answer.

To evaluate the effectiveness, we implement ChatCoT on ChatGPT, and conduct experiments on two complex reasoning benchmarks, *i.e.,* MATH (Hendrycks et al., 2021) and HotpotQA (Yang et al., 2018). Experimental results show that ChatCoT achieves very promising performance on MATH with 7.9% relative improvement in average over the SOTA baselines (*i.e.,* PHP (Zheng et al., 2023)). Besides, our approach can also be integrated with other strategies, *e.g.,* self-consistency, and ChatCoT can achieve better performance by incorporating these strategies.

## 2 Related Work

**Tool-Augmented Large Language Models.** With the large-scale parameters and pre-training corpus, large language models (LLMs) (*e.g.,* Flan T5 (Chung et al., 2022), ChatGPT (OpenAI, 2022) and LLaMA (Touvron et al., 2023)) have demonstrated strong zero-shot and few-shot ability in NLP tasks (*e.g.,* language generation, reasoning). However, LLMs have still struggled with complex reasoning tasks requiring task-specific knowledge and multi-step reasoning (*e.g.,* mathemati-

cal problem solving). Previous work (Zhao et al., 2022, 2023b; Luo et al., 2023) has constructed task-specific corpus and utilized continue pre-training and instruction tuning to inject relative knowledge into LLMs and enhance the complex reasoning ability of LLMs. In order to further reduce the mistakes made by LLMs, existing methods have explored to augment LLMs with external tools. They can be roughly categorized into the following two types. The first type of methods (Gao et al., 2023; Parisi et al., 2022; Qiao et al., 2023) train the model parameters to support the utilization of the external tools, where they collect or synthesize the tool-augmented examples to tune the model parameters (Schick et al., 2023; Patil et al., 2023; Hao et al., 2023). Another type of methods (Gao et al., 2022; Yao et al., 2022; Zhang et al., 2023) utilize carefully designed prompts to guide LLMs to use external tools. They focus on devising proper prompts or tools manipulation ways to select and use tools when necessary (Liang et al., 2023; Shen et al., 2023; Yao et al., 2022). In this work, we follow the second type of methods and propose a tool-augmented chain-of-thought reasoning strategy that can better solve complex reasoning tasks.

**Chain-of-Thought Reasoning.** To further enhance the reasoning capacity of LLMs, Chain-of-Thought (CoT) prompting strategy (Wei et al., 2022; Kojima et al., 2022) has been proposed to guide LLMs to generate intermediate reasoning steps which can boost the performance of LLMs. Through special instructions (*e.g., "Let us think step by step"*) and in-context exemplars with detailed intermediate reasoning steps, LLMs can perform step-by-step reasoning to reach the final answer. Based on CoT, recent work has also proposed several methods to further improve the performance, including problem decomposition (Zhou et al., 2022; Dua et al., 2022), appropriate exemplars selection (Ye et al., 2022; Shi et al., 2023), results post-processing (Wang et al., 2022; Madaan et al., 2023; Zheng et al., 2023), and changing the reasoning format (Yao et al., 2023; Wu et al., 2023). However, as the generation process of CoT is one-pass, the utilization of tools in intermediate steps would have to interpret it, hurting the continuity of the generation process. In this work, we propose a unified way to integrate CoT reasoning and tool manipulation, which utilizes the excellent multi-turn chatting capacity of LLMs to perform CoT reasoning across multi-turn conversations.

## 3 Preliminary

In this section, we present the task setting, then introduce the Chain-of-Though prompting strategy and the tool manipulation in reasoning tasks.

**Task Setting.** In this work, we focus on improving the reasoning ability of large language models (LLMs) on complex tasks, *e.g.,* solving mathematics competition problems. Unlike tasks that can be solved by humans via straightforward skills or tools, complex tasks require advanced knowledge (*e.g.,* mathematical theorem) and multi-step reasoning to reach the answer. Typically, a complex problem includes three types of texts, namely problem statement, solution text, and answer key, denoted as $Q$, $S$ and $A$, respectively. The problem statement $Q$ introduces the background and description of a complex problem, and the solution text illustrates the detailed solving process to obtain the answer key. All of them are composed of a sequence of tokens, where each token is either a text word or a mathematical symbol. Formally, given the problem statement $Q$, we aim to utilize LLMs to perform multi-step reasoning, to finally generate its accurate answer $A$.

**Chain-of-Thought Prompting.** To elicit the powerful reasoning ability of LLMs for complex tasks, Chain-of-Thought (CoT) prompt strategy (Wei et al., 2022) has been widely used to guide LLMs for performing step-by-step reasoning. Generally, the CoT prompt consists of few exemplars whose a series of intermediate reasoning steps $\{I_1, \cdots, I_n\}$ are also involved. Each exemplar can be denoted as $E = \langle Q, \{I_1, \cdots, I_n\}, A \rangle$. Formally, given the question and few exemplars, a CoT prompt is composed by integrating them as a long input of the LLM, which can prompt the LLM to generate a similar thought chain that leads to the final answer.

**Tool Manipulation.** Previous work has revealed that LLMs are struggling with basic functionality (*e.g.,* arithmetical calculation (Schick et al., 2023)), which can be solved by using specific external tools (*e.g.,* calculator), denoted as $\{T_1, \ldots, T_n\}$. To manipulate tools, existing work mostly relies on writing a detailed prompt for describing how to use available tools for the LLM, then incorporates it to guide the selection of useful tools and generate the tool arguments, and finally calls the tool API to obtain the result. Following this way, in this work, we focus on three useful tools that have been widely

used by humans to solve complex problems:

- *Calculator:* Given a mathematical expression, the calculator can compute the value of it or simplify it according to arithmetic rules (*e.g.,* combining like terms and reduction of fractions).

- *Equation Solver:* Given the equations system and unknown variables, the equation solver can automatically calculate the value of the contained unknown variables through relative algorithms.

- *Retriever:* Given a query, the retriever aims to extract the most relevant information (*e.g.,* documents) from a number of candidates. According to the types of the retrieved corpus, it can be implemented by specialized models, *e.g.,* dense retrieval model.

We implement the first two tools by using different functions of *SymPy* (Meurer et al., 2017), a Python library for mathematical symbolic calculation. For the retriever, we adopt SimCSE (Gao et al., 2021), a sentence embedding model to measure the text semantic similarity. Note that when the input expression or equation is ill-formed or unsolved, the above tools would return an error.

## 4 Approach

In this section, we present our proposed ChatCoT, a new chain-of-thought (CoT) prompting framework based on multi-turn conversations, for improving chat-based LLMs on complex reasoning tasks with tools. The overall illustration of our proposed Chat-CoT is shown in Figure 1.

### 4.1 Overview

For complex tasks (*e.g.,* advanced mathematical problems), LLMs require to frequently manipulate the tools when in need, to fulfill the intractable intermediate issues. However, as tools are not intrinsically integrated with LLMs, previous work mostly relies on the LLM to generate the plan of manipulating tools and then execution (Gao et al., 2022; Lu et al., 2023), or immediately call tools by stopping the continuous generation of LLMs (Yao et al., 2022). The above both ways are not suitable for the frequent interactions between LLMs and tools, due to the error accumulation in planning and frequent interruptions in LLM generation.

In our approach, we decompose the chain-of-thought reasoning process of LLMs into a multi-round conversation. In each turn, LLMs just require to concentrate on manipulating tools or accomplishing reasoning in the current step, and the whole

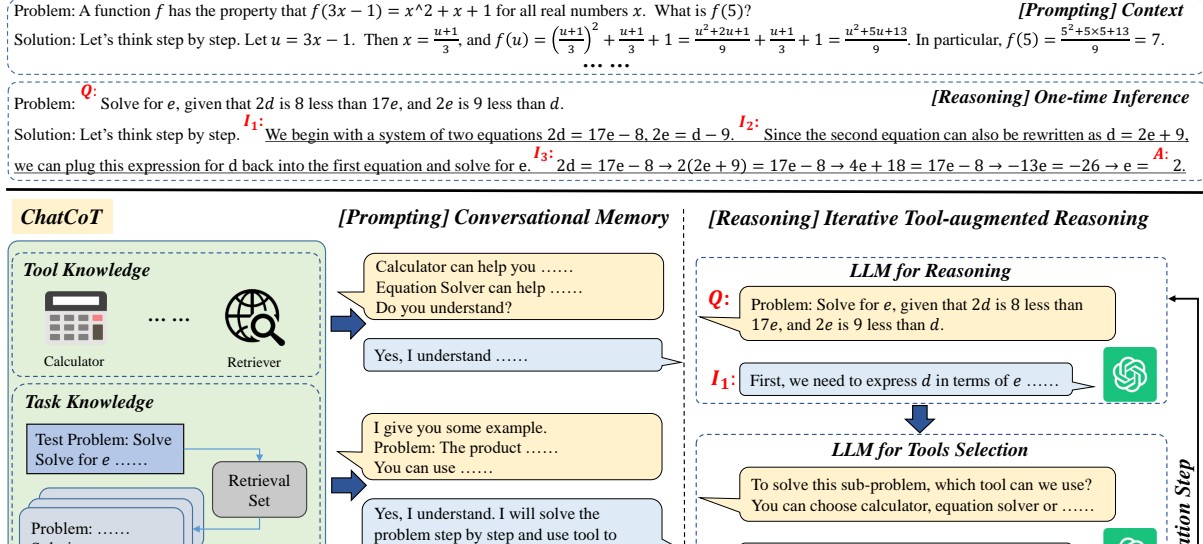

Figure 1: The comparison of vanilla CoT and ChatCoT, illustrated for a mathematical problem. For vanilla CoT, the content underlined are generated by LLMs. For ChatCoT, the conversational knowledge memory is initialized to provide tools, task and reasoning format knowledge. Then, the tool-augmented reasoning step is iterated multiple times to perform step-by-step reasoning, until obtaining the answer.

reasoning process would keep on pushing without premature planning and sudden interruption. In this way, the whole reasoning process would be converted to a conversation between LLMs and an agent, which follows pre-defined rules to guide LLMs and manipulate the tool. By designing proper chatting strategies, the agent would automatically elicit LLMs to perform reasoning and select a tool, or invoke the tool for execution.

In our approach, we first initialize the multi-turn conversation by feeding chat-based LLMs with the background knowledge, *i.e.,* the description of tools, relevant task exemplars, and the demonstration of decomposed chain-of-thought in chat, which are the *conversational knowledge memory* for supporting the following reasoning. Then, we propose the *tool-augmented reasoning* procedure that leverages LLMs to perform reasoning with tools in the current step and iterate it to fulfill all sub-tasks in the whole reasoning process, until reaching the answer. We introduce the details of the two components in the following.

## 4.2 Initializing Conversational Knowledge Memory

To guide chat-based LLMs to follow our proposed ChatCoT using external tools, it is essential to design proper prompts in context. In our approach, as we reformulate the chain-of-thought reasoning into a decomposed multi-turn conversation, we can also feed the essential prompts into LLMs at early turns as the context, to initialize the conversation background knowledge. It can be seen as the in-context *knowledge memory* in the format of dialogue that stores useful knowledge for helping chat-based LLMs manipulate tools or perform reasoning. Here, we consider three types of knowledge about tools, task, and multi-turn reasoning format, respectively. The details of prompts are in Appendix A.

**Tools Knowledge.** As LLMs have never seen tools during pre-training, for each tool in Section 3, we hand-craft its description in the following pattern: "*[T] can help you [Y]*", where *[T]* is the tool name and *[Y]* shows its detailed functionality. Then, we merge all the descriptions and design the input

prompt to tell LLMs about the knowledge of all tools. We also hand-craft the expected response of the LLM. It will be also fed into the LLM, to indicate the LLM that it has accepted our prompt and should follow it.

**Retrieval-Augmented Task Knowledge.** Since LLMs can learn the task knowledge from in-context exemplars, we leverage a retriever to select the most relevant instance from the training dataset, to provide more useful knowledge for the given question. Concretely, we train SimCSE (Gao et al., 2021), a sentence embedding method that can measure the semantic similarity of texts, via the unsupervised training strategy on the training set. Then, we leverage it to retrieve the top-$k$ most semantically similar exemplars, and concatenate their problem statement $Q$ and solution $S$ to compose the input prompt. Similarly, we also feed it with our expected response into the LLM.

**Multi-turn Reasoning Format.** To elicit LLMs following multi-turn reasoning format, we manually annotate the whole multi-round dialogue $I_1, \cdots, I_n$ of randomly sampled five questions from the training set, to create the exemplars. Then, we feed the dialogues of all the exemplars into the chat-based LLM round by round, as the context to guide LLMs to follow it for performing reasoning.

**Summary.** The above three types of multi-turn utterances are pre-defined with corresponding contents and formats, which compose the *conversational knowledge memory* of our approach. It would be leveraged to initialize the conversational context, and support the following step-by-step reasoning for answering the question.

## 4.3 Iterative Tool-augmented Reasoning

Based on the above conversational knowledge memory, we iterate the tool-augmented reasoning step to perform step-by-step tool-augmented reasoning, until finally obtain the answer.

### 4.3.1 Tool-augmented Reasoning Step

The tool-augmented reasoning step can be iterated in multiple times. In each iteration, based on the current results, we first leverage LLMs to perform reasoning, then select the proper tool by LLMs, and finally execute the selected tool to obtain the intermediate result in the current step.

**LLM for Reasoning.** Guided by the exemplars in the conversation history, LLMs are able to decom-

pose the whole reasoning process into multi-turn chat. Specially, LLMs would be elicited by the contextual exemplars to directly perform reasoning in natural language based on the current result, without specialized prompts or instructions. Consequently, LLMs can rely on the retrieval-augmented task knowledge in context, to generate the natural language solution till the point that needs the functionality of tools.

**LLM for Tools Selection.** After reasoning, we utilize the LLM to select a useful tool (*e.g.,* calculator), which will be employed to provide the required functionality for the LLM. Here, the input prompt of the LLM is "*To solve this sub-problem, which tool can we use?*" After feeding it into the LLM, if the LLM requires to utilize tools, it will select a suitable one, and then we further ask the LLM to formulate the input arguments of the tool, *e.g.,* mathematical expression. Otherwise, it will answer "*Do not use tool*", and the LLM will continue to perform reasoning.

**Tools Execution.** Given the selected tool and formulated arguments by LLMs, we can execute the tool with the arguments to obtain the result in the current iteration. Here, we also consider that the results from the tool may be not satisfied by the LLM, *e.g.,* irrelevant retrieved documents. In this case, we can also add several feedback rounds where the LLM judges if the result is useful or expected, and then reuse the tool to acquire a new result.

### 4.3.2 Iteration for Step-by-Step Reasoning

We iterate the above step based on the in-context conversation knowledge memory, to perform step-by-step reasoning on the given question $Q$. We start the whole iteration process using the following prompt: "*You should solve the problem step by step and you should follow the react in the history [Q]*". Then, after reaching the answer key, the iteration process will be stopped by LLMs. In practice, we find that chat-based LLMs are prone to continue chatting although the answer key has appeared in the reasoning process. Thus, we set the maximum chat turns, and devise the following prompt to force LLMs to stop reasoning and conclude the answer: "*Base on the context, what is the answer?*".

As our proposed approach only decomposes the one-pass chain-of-thought reasoning into multi-turn chat and adds the utilization of tools, it is agnostic to the task types and tools implementation. Therefore, it is a general framework that can

| Dataset | Category | Train | Dev/Test |
|---------|----------|-------|----------|
| MATH | Algebra | 1744 | 1187 |
| | CP | 771 | 474 |
| | Precalculus | 746 | 546 |
| | Prealgebra | 1205 | 871 |
| | Geometry | 870 | 479 |
| | IA | 1295 | 903 |
| | NT | 869 | 540 |
| HotpotQA | Distractor | 90477 | 7405 |

Table 1: Statistics of the two complex reasoning datasets. CP, IA, and NT denote *Counting and Probability*, *Intermediate Algebra*, and *Number Theory*, respectively.

be applied to a variety of complex reasoning tasks that require suitable tools. Besides, our approach also supports the recently proposed improvement strategies based on the chain-of-thought method, *e.g.,* self-consistency (Wang et al., 2022). We conduct corresponding experiments in Section 5.3 to validate it.

## 5 Experiment

In this section, we conduct experiments to evaluate the effectiveness of ChatCoT. The implementation details can be found in Appendix B.

### 5.1 Experimental settings

**Datasets.** We consider two complex reasoning datasets for evaluation, *i.e.,* MATH (Hendrycks et al., 2021) and HotpotQA (Yang et al., 2018). The details of these two datasets are shown in Table 1. We adopt accuracy as the evaluation metric.

• *MATH* is composed of challenging competition mathematical problems which require advanced mathematical knowledge. It is divided into seven categories, *i.e.,* Algebra, Counting and Probability, Precalculus, Prealgebra, Geometry, Intermediate Algebra, and Number Theory. We adopt the calculator and an equation solver as external tools to help LLMs.

• *HotpotQA* is a multi-hop question answering dataset, where each question is associated with a collection of paragraph candidates containing several golden contents which are useful for reasoning. We use the development set under the distractor setting of HotpotQA for evaluation, where the annotation of golden paragraphs is not aware to LLMs. We employ the retriever as the external tool.

**Baselines.** We mainly compare our approach with

the following prompting strategies based on Chat-GPT (OpenAI, 2022):

• *Chain-of-Thought (CoT)* (Wei et al., 2022) is a prominent method to boost the performance of LLMs in reasoning tasks. In CoT, LLMs are prompted to generate the intermediate reasoning path and reasoning step by step to reach the final answer. Previous work has shown that the utilization of external tools and similar exemplars improves the performance of CoT. Therefore, we implement external tools to help LLMs reason and retrieve to help LLMs select exemplars, which are named *CoT w/ Tool*, and *CoT w/ Retri*, respectively.

• *Learning-to-Program (LP)* (Guo et al., 2023) guides LLMs to program in natural language by learning solutions in the training set, and elicits LLMs to solve tasks following the program.

• *Progressive-Hint Prompting (PHP)* (Zheng et al., 2023) proposes to iteratively refine the solution based on the answer hints from previous trials. The iterative method achieves SOTA on MATH.

To provide a more complete evaluation, we also report the performance of various LLM backbones with the vanilla CoT prompting, including PaLM (Chowdhery et al., 2022), PaLM 2 (Google, 2023), Minerva (Lewkowycz et al., 2022), Galactica (Taylor et al., 2022), LLaMA (Touvron et al., 2023) and GPT-3 (Brown et al., 2020).

### 5.2 Main Results

We present the evaluation results of our approach on MATH and HotpotQA datasets in Table 2 and Table 3 respectively.

First, for the comparison of backbones for CoT prompting, ChatGPT achieves the best performance, demonstrating its outstanding mathematical reasoning ability. Our method elicits the reasoning process by leveraging the strong multi-turn dialogue ability of ChatGPT, thus leading to a better release of the reasoning ability from ChatGPT.

Second, retrieval-augmented methods (*e.g.,* ChatCoT, CoT w/ Retri) outperform other baselines. The reason is that retrieved exemplars may contain more relevant knowledge and reasoning steps that are beneficial to solve the given problem. On Geometry tasks of MATH, CoT w/ Retri achieves the largest improvement over vanilla CoT than other sub-tasks. Another possible reason is that ChatGPT is more unfamiliar to the knowledge and symbol of geometry than others. Without similar exemplars, it is difficult for LLMs to well understand them.

| Models | Prompt Strategy | MATH | | | | | | | |
|---|---|---|---|---|---|---|---|---|---|
| | | Algebra | CP | PC | PA | Geometry | IA | NT | Avg. |
| GPT-3 | CoT | 6.0 | 4.7 | 4.0 | 7.7 | 3.1 | 4.4 | 4.4 | 5.2 |
| PaLM | CoT | 9.7 | 8.4 | 4.4 | 19.2 | 7.3 | 3.5 | 6.0 | 8.8 |
| LLaMA | CoT | - | - | - | - | - | - | - | 10.6 |
| Galactica | CoT | 29.0 | 13.9 | 12.8 | 27.2 | 12.3 | 9.6 | 11.7 | 20.4 |
| Minerva | CoT | 51.3 | 28.0 | 18.0 | 55.0 | 26.8 | 13.7 | 21.2 | 33.6 |
| PaLM 2 | CoT | - | - | - | - | - | - | - | 34.3 |
| ChatGPT | CoT | 48.1 | 31.4 | _21.1_ | 56.6 | 22.3 | 18.3 | 29.1 | 35.1 |
| | CoT w/ Tool | 35.9 | 22.6 | 9.3 | 40.5 | 13.6 | 9.4 | 19.4 | 23.8 |
| | CoT w/ Retri | _52.7_ | 32.7 | 18.9 | _58.4_ | _29.2_ | **19.9** | 31.7 | _37.7_ |
| | LP | 49.6 | 30.2 | 16.3 | 52.3 | 22.5 | 16.9 | 29.8 | 34.0 |
| | PHP | 51.1 | _33.7_ | 16.1 | 57.7 | 25.4 | 17.1 | **35.1** | 36.5 |
| | ChatCoT | **56.1** | **34.2** | **23.8** | **59.2** | **29.9** | _19.5_ | _32.6_ | **39.4** |

Table 2: Experimental results on MATH dataset. PC and PA denote *Precalculus* and *Prealgebra*, respectively. Avg. is the average value of all categories. The best are denoted in bold and the second-best are underlined.

| Methods | HotpotQA |
|---|---|
| CoT | 38.0 |
| CoT w/ Tool | 31.4 |
| ChatCoT w/o Feedback | _53.8_ |
| ChatCoT | **59.2** |

Table 3: The results on HotpotQA. We report the results of the development set under the distractor setting.

| Methods | | | MATH | | |
|---|---|---|---|---|---|
| TK | RATK | MRF | PC | Geo | NT |
| ✔ | ✔ | ✔ | **23.8** | **29.9** | **32.6** |
| ✗ | ✔ | ✔ | 23.3 | 29.2 | 30.6 |
| ✔ | ✗ | ✔ | 20.0 | 27.4 | 31.0 |
| ✔ | ✔ | ✗ | 21.6 | 24.2 | 32.2 |
| ✗ | ✗ | ✔ | 16.7 | 21.1 | 29.3 |

Table 4: The results of ablation study. TK, RATK, and MRF denote if using tool knowledge, retrieval-augmented task knowledge, and multi-turn reasoning format at early turns of the conversation, respectively. Geo is the abbreviation of *Geometry*.

Third, given the results of CoT and CoT w/ Tool on MATH and HotpotQA, we can find that directly utilizing external tools during reasoning is not a suitable way, which may hurt the performance of LLMs. The reason may be that injecting tool usage into the CoT reasoning process will hurt the continuity of reasoning.

Finally, ChatCoT achieves state-of-the-art performance on MATH dataset based on ChatGPT and outperforms other baselines on HotpotQA. Compared with the previous SOTA method PHP, ChatCoT outperforms six of seven sub-tasks on MATH dataset and achieves 7.9% relative improvement on average accuracy over the PHP method. The experiment results have verified the effectiveness of ChatCoT on complex reasoning tasks. By leveraging conversational knowledge memory and multi-round dialogue to reasoning, ChatCoT has the advantage to utilize plug-and-play tools. Moreover, on the Number Theory tasks of MATH, we can find that PHP achieves the best performance. The reason may be that there are fewer equations that need to be computed or simplified. Thus, the advantage

of the utilization of tools becomes less obvious.

### 5.3 Detailed Analysis

In order to further verify the effectiveness of each component in ChatCoT, we conduct experiments about ablation, adaption, tools utilization and expense. We present the case study in Appendix C.1.

**Ablation Study.** In the ablation study, we evaluate the effectiveness of conversational memory, including tool knowledge memory, retrieval-augmented knowledge memory, and multi-turn reasoning format memory. As shown in Table 4, removing any type of conversational memory will reduce the performance of ChatCoT, which indicates the effectiveness of these memories in complex reasoning. In particular, removing retrieval-augmented knowledge memory or multi-turn reasoning format memory will lead to a large drop, which shows

| Methods | CP | NT |
|---|---|---|
| CoT + SC | $35.2_{+3.8\%}$ | $34.4_{+5.3\%}$ |
| ChatCoT + SC | $40.1_{+5.9\%}$ | $38.3_{+5.7\%}$ |

Table 5: The evaluated accuracy of combining our approach with self-consistency. SC denotes self-consistency. We also report the absolute improvement compared with vanilla methods on subscripts.

| Methods | Frequency | Success |
|---|---|---|
| CoT w/ Tool | 3.0% | 85.7% |
| ChatCoT w/o TK | 56.0% | 93.0% |
| ChatCoT w/o MRF | 10.0% | 64.2% |
| ChatCoT | 70.0% | 92.0% |

Table 6: Frequency and success rate of tool manipulation on Number Theory task of MATH. TK, MRF denote tool knowledge, multi-turn reasoning format at early turns of the conversation respectively.

that mathematical knowledge and reasoning format knowledge is important for LLMs in reasoning tasks, while LLMs can learn the usage of external tools from exemplars without descriptions.

**Combination with Improvement Strategies.** ChatCoT is a general method to enhance the ability of tool manipulation of LLMs. It can be integrated with improvement strategies and further boost the performance of LLMs on reasoning tasks. To evaluate the applicability of ChatCoT to improvement strategies designed for CoT, we compare ChatCoT with CoT on two subtasks of MATH, where both methods are augmented with self-consistency (Wang et al., 2022), a representative improvement strategy for CoT prompting. Concretely, we sample 5 outputs for majority voting in self-consistency. As shown in Table 5, self-consistency brings improvement in both CoT and ChatCoT. In particular, the absolute improvement of ChatCoT is slightly higher than CoT, showing that ChatCoT can adapt to self-consistency well. The reason is that, with the decomposing of reasoning procedures, the intermediate steps of ChatCoT are more confident, and small mistakes will be corrected easily. Moreover, we construct the case study about the combination with ChatCoT and Self-Refine (Madaan et al., 2023) in Appendix C.2.

**Tools Utilization Analysis.** As we mentioned above, in complex reasoning tasks, infrequently or incorrectly utilizing external tools might lead to wrong answers. Thus, we conduct the experiment

| Methods | Generated Tokens |
|---|---|
| CoT | 224.6 |
| CoT w/ Tool | 296.2 |
| Self-Consistency | 1017.4 |
| ChatCoT | 355.2 |

Table 7: The comparison of the number of generated tokens from LLMs among different prompt strategies.

about whether LLMs can frequently or correctly leverage based on different methods. Table 6 expresses the performance of tools utilization in the Number Theory task of MATH of baseline and our approach. "Frequency" denotes the ratio of problems where LLMs correctly leverage tools. "Success" denotes the rate of LLMs utilizing tools successfully among all the times of invoking tools. We can observe that ChatCoT achieves a balance of frequency and ratio of success. Tool knowledge provides the function of tools for LLMs and improves the frequency that LLMs utilize the tools. LLMs can learn how to leverage external tools through the multi-turn reasoning format and boost the ratio of successful utilization of tools. Without any of them, the frequency and ratio of success will drop which might not be conducive to reasoning.

**Number of Generated Tokens Analysis.** Despite guiding LLMs to reason through multi-turn dialogue, the computation expense of ChatCoT is not significantly larger than the CoT method. In Table 7, we present the average number of generated tokens from LLMs in several methods on MATH dataset, which reflects the computation expense to a certain degree. We can observe that ChatCoT is on the same order of magnitude with other baselines (*e.g.,* CoT and CoT w/ Tool). Therefore, ChatCoT does not bring significant expenses compared to existing prompting methods.

## 6 Conclusion

In this paper, we have proposed ChatCoT, a new framework to manipulate the tools for the CoT reasoning. It naturally integrates the reasoning process and manipulating tools through a form of multi-turn conversations. At each turn, LLMs can either interact with tools or perform the reasoning by itself. Our approach can effectively leverage the multi-turn conversation ability of chat-based LLMs. Experimental results on two complex reasoning tasks including MATH and HotpotQA have verified the

effectiveness of ChatCoT.

Currently, our experiments are mainly conducted on mathematical reasoning tasks, and we will test the effectiveness of the proposed approach to more types of reasoning tasks. Besides, we will also consider extending the number of available tools for solving different tasks.

## Limitations

In this section, we discuss the limitations of our work. First, we do not utilize GPT-4 in our experiment or evaluate the performance of GPT-4 in the ChatCoT framework. That is because our application for GPT-4 has not been accepted. Second, ChatCoT is designed for chat-based LLMs and it is hardly compatible with other LLMs. However, most LLMs support multi-turn conversation currently and they perform well on reasoning tasks. Besides, although LLMs have achieved strong ability in reasoning tasks, the requirement of computation expense and GPU resource is higher than other pre-trained language models which have millions of parameters. The utilization of LLMs will produce more carbon dioxide and pollute the environment.

## Acknowledgement

This work was partially supported by National Natural Science Foundation of China under Grant No. 62222215, Beijing Natural Science Foundation under Grant No. L233008 and 4222027, and Beijing Outstanding Young Scientist Program under Grant No. BJJWZYJH012019100020098. And this work is also partially supported by the Outstanding Innovative Talents Cultivation Funded Programs 2021 of Renmin University of China. Xin Zhao is the corresponding author.

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

## A  Details of Conversation Memory

In this part, we present the details of the prompt in conversation Memory.

**Tool Knowledge.** The two turns of utterances are:

*User:* "*You can use tool to help you solve the problem and I give you the instruction of tools usage. $[T_1]$ can help you $[Y_1] \cdots$ Do you understand?*"

*LLM:* "*Yes, I understand. I will use tool to help me solve the problem.*".

**Retrieval-Augmented Task Knowledge.** The two-turn utterances are:

*User:* "*I give you some example. Problem: $[Q_1]$ Solution: $[S_1] \cdots$ You can use the knowledge and thoery in these problem. Do you understand?*"

*LLM:* "*Yes, I understand. I will solve the problem step by step and use tool to help me.*".

**Multi-turn Reasoning Format.** The multi-turn utterances are based on the following pattern:

*User:* "*Problem: $[Q'_1]$ Let's think step by step and use knowledge in similar problem to solve this problem.*"

*LLM:* "*$[I_1]$*".

...

*LLM:* "*$[I_n]$*".

## B  Implementation Details

During the evaluation, we utilize ChatGPT (gpt-3.5-turbo) (OpenAI, 2022) as our backbone model, and fine-tune RoBERTa (Liu et al., 2019) following SimCSE (Gao et al., 2021) on the training sets of MATH and HotpotQA separately as the retriever in corresponding tasks.

For MATH, we leverage 5-shot setting. The exemplars of CoT and CoT w/ Tool are randomly sampled, while exemplars of CoT w/ Retri are retrieved top-5 similar problems by the retriever. For ChatCoT, 2 retrieval exemplars and 3 annotated exemplars will be adopted. For HotpotQA, we leverage 4-shot setting which is similar to MATH, due to the length limitation of input. For the CoT method, we retrieve the top-3 relevant paragraphs from the paragraph collection as evidence of the given question. In ChatCoT, as the retrieved paragraphs might be not useful for LLMs, LLMs can send feedback to the retriever to show other results at most 5 times.

## C  Case Study

### C.1  Framework of ChatCoT

In order to better present the reasoning process of ChatCoT, we conduct the case study of two problems in MATH dataset, which is shown in Figure 2.

The beginning prompt contains knowledge of tools, tasks, and reasoning format. In the tool knowledge, we introduce the usage and function of external tools. For task knowledge, we retrieve similar problems and corresponding solutions from the training set as retrieval exemplars through semantics similarity, which might contain relevant knowledge. The reasoning format is used to guide LLMs to reason through multi-round dialogue. The beginning prompt and corresponding response are pre-set to enhance the sensitivity of LLMs to instructions.

Instructed by the beginning prompt, LLMs iterate the tool-augmented reasoning step for step-by-step reasoning in ChatCoT. We illustrate the flow of ChatCoT with the example on the left in Figure 2. Given the math problem, LLMs first reason out that *d should be expressed in terms of e*. Then, LLMs communicate with the agent to select the appropriate external tool *Equation Solver* and specify the corresponding arguments to invoke the tool, including *the unknown symbols* and *the system of equations*. Based on the selection, The assistant executes the tool invocation and returns *the solution of the system of equations* to LLMs. After execution, LLMs can perform the next reasoning step based on the given result. The iterative tool-augmented reasoning process can be iterated several times until LLMs find out the answer. In this case, LLMs find that the original problem has been solved, thus obtaining the final answer and ending the reasoning process.

### C.2  Combination with Self-Refine

To better show how LLMs correct mistakes through ChatCoT, we present a case study of the combination of ChatCoT and Self-Refine in the following:

*User*: "*Problem: What is the remainder when $13^{13} + 5$ is divided by 6?*

*Let's think step by step*"

*LLM*: *First, we need to find out the value of $13^{13} + 5$ using the properties of exponents.*

......

*LLM*: "*Second, we need to find out the remainder when 302875106592258 is divided by 6.*"

**User**: "To solve this sub-problem, which tool can we use?"

**LLM**: "*Calculator*"

**User**: "*Give me the equation to calculate*"

**LLM**: "302875106592258  mod 6"

**User**: "*Results:* 302875106592258  mod 6 = $1.81725063955355e + 15 * mod.$

*Continue reasoning*"

**LLM**: "*Finally, we know that the remainder when $13^{13} + 5$ is divided by 6 is the same as the remainder when 302875106592258 is divided by 6, which is* $\boxed{2}$"

**User**: "*The solution above might some mistake, you should check the solution and get the final answer.*"

**LLM**: "*Apologies for the mistake. Let's correct the solution.*

*To find the remainder when $13^{13} + 5$ is divided by 6, we can simplify the expression $13^{13}$ first.*

*Using the property of remainders, we can observe that $13 \equiv 1 \pmod 6$. Therefore, $13^{13} \equiv 1^{13} \equiv 1 \pmod 6$.*

*Now, we can rewrite the expression as $13^{13} + 5 \equiv 1 + 5 \equiv 6 \equiv \boxed{0} \pmod 6$.*

*Hence, the remainder when $13^{13} + 5$ is divided by 6 is 0.*"

In this example, we can observe that LLMs try to calculate the exact value of $13^{13}+5$ at first, and then get the remainder. However, the calculator does not understand the expression "302875106592258 mod 6" and returns the wrong result. Through refining, LLM changes the solution and gets the correct answer.

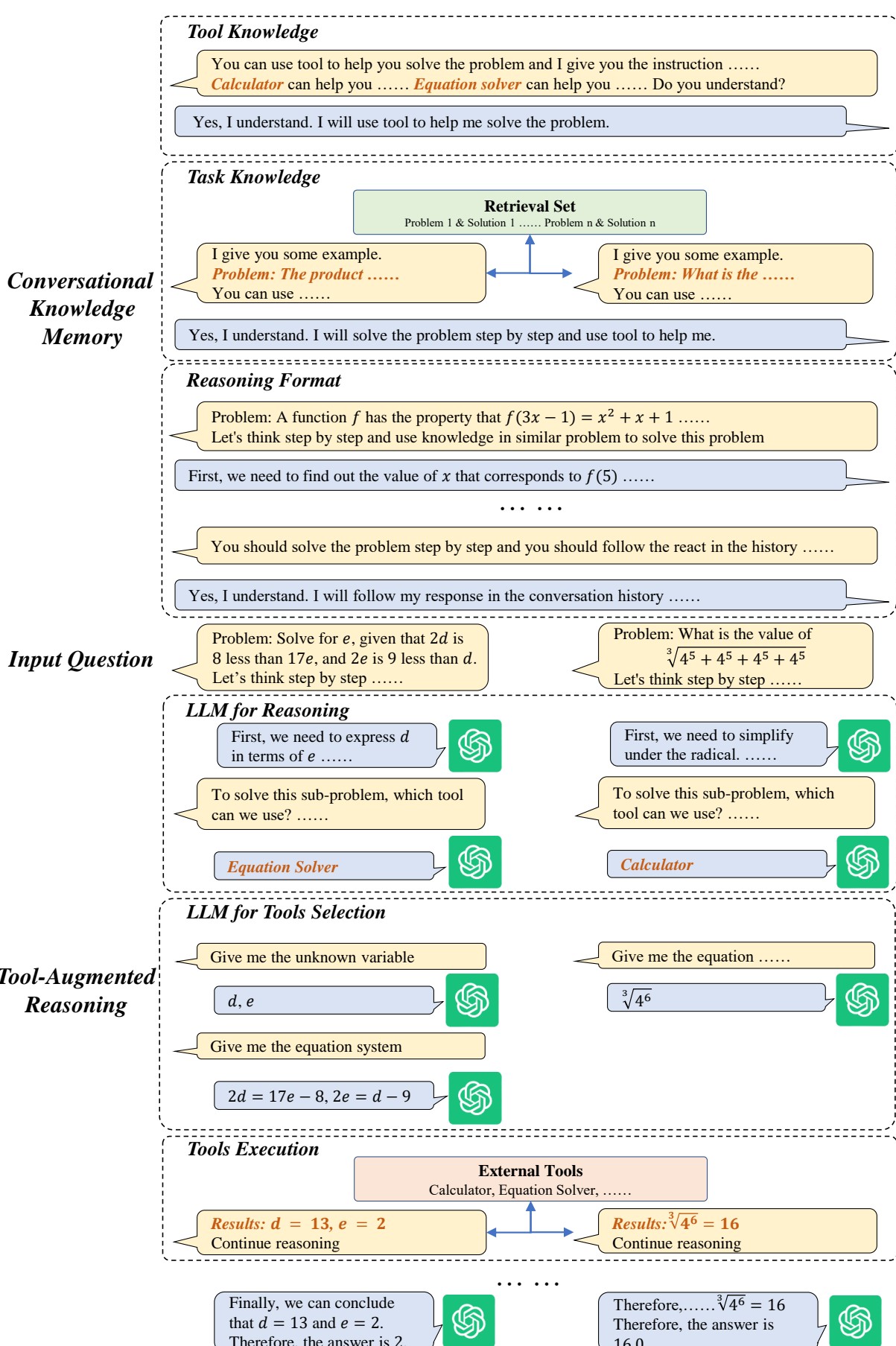

Figure 2: An illustration example for ChatCoT from MATH.