# OpenReview forum: "ChatCoT: Tool-Augmented Chain-of-Thought Reasoning on Chat-based Large Language Models"
_EMNLP/2023/Conference — EMNLP 2023 Findings_

### Official Review · Reviewer_iNwF · 2023-08-01

**Soundness:** 3

**Excitement:**

2: Mediocre: This paper makes marginal contributions (vs non-contemporaneous work), so I would rather not see it in the conference.

**Paper Topic And Main Contributions:**

The paper introduces ChatCoT, a tool-augmented chain-of-thought reasoning framework designed to enhance the reasoning abilities of chat-based large language models (LLMs) such as ChatGPT. ChatCoT models the chain-of-thought (CoT) reasoning as multi-turn conversations, enabling LLMs to interact with tools and perform reasoning in a unified manner. The proposed approach effectively leverages the multi-turn conversation ability of chat-based LLMs and achieves a 7.9% relative improvement over the state-of-the-art baseline on complex reasoning tasks in two datasets (MATH and HotpotQA).

Advantages:
The paper proposes ChatCoT, a tool-augmented chain-of-thought reasoning framework, to enhance the reasoning capabilities of chat-based LLMs, providing valuable insights for improving complex reasoning tasks.
The writing style is straightforward and comprehensible, facilitating understanding of the proposed framework.

Disadvantages:
The novelty of the proposed method is relatively weak, as it does not substantially deviate from existing approaches.
In the experimental evaluation, the authors should compare ChatCoT not only with the simplest usage of LLMs but also with other large-model-based methods. Additionally, a comparison with fine-tuned smaller models would be useful.

**Reasons To Accept:**

The paper proposes ChatCoT, a tool-augmented chain-of-thought reasoning framework, to enhance the reasoning capabilities of chat-based LLMs, providing valuable insights for improving complex reasoning tasks.

The writing style is straightforward and comprehensible, facilitating understanding of the proposed framework.

**Reasons To Reject:**

The novelty of the proposed method is relatively weak, as it does not substantially deviate from existing approaches.

In the experimental evaluation, the authors should compare ChatCoT not only with the simplest usage of LLMs but also with other large-model-based methods. Additionally, a comparison with fine-tuned smaller models would be useful.

**Reproducibility:**

3: Could reproduce the results with some difficulty. The settings of parameters are underspecified or subjectively determined; the training/evaluation data are not widely available.

**Reviewer Confidence:**

3: Pretty sure, but there's a chance I missed something. Although I have a good feel for this area in general, I did not carefully check the paper's details, e.g., the math, experimental design, or novelty.

---

> ### Author Rebuttal · Authors · 2023-08-28
>
> We sincerely thank the reviewer for the insightful suggestion.
>
> Our approach involves synergizing the chain-of-thought (CoT) prompting strategy with frequent tool manipulation operations to enhance the reasoning prowess of LLMs, while maintaining the uninterrupted flow of CoT reasoning. The primary hurdle in achieving this lies in upholding the coherence of the CoT reasoning process post-tool integration. However, the single-pass generation approach of the conventional CoT method doesn't facilitate this objective. Hence, we propose a transition to a multi-turn conversational approach, wherein each utterance represents an intermediate step in the overarching reasoning process. This modification allows for the seamless introduction of tool invocations as needed, after each utterance.
>
> Additionally, within our methodology, we introduce other innovative strategies to enhance its performance even further. Initially, since the comprehensive reasoning procedure aligns with a multi-turn conversational approach, we introduce the conversational memory mechanism. This mechanism offers valuable insights to LLMs during the initial phases of the conversation, encompassing tool knowledge, retrieval-augmented task knowledge, and the multi-turn reasoning structure. Secondly, we meticulously craft prompts that direct LLMs to effectively utilize the information stored in the conversational memory. This empowers them to engage in an iterative process of reasoning with tools, persisting until the solution is reached.
>
> For the purpose of delineating the distinctions between our approach and established methods, we also broadly classify prior research on tool enhancement into the subsequent two categories. In the ensuing discussion, we provide a succinct overview of each category and subsequently outline the disparities between our approach and each of these categories.
> 1. Previous work has adopted the planning-then-execution paradigm to enhance the reasoning acumen of LLMs. In such kind of work (e.g., PAL [1]), LLMs are tasked with formulating a comprehensive plan for the entire reasoning procedure, encompassing the strategic arrangement of tools for manipulation. Subsequently, this preconceived procedure is executed to finalize the reasoning process, with no room for LLMs to engage with tools or modify the plan post-generation, despite glaring errors. In contrast, our approach affords LLMs the flexibility to seamlessly interact with tools throughout the entire reasoning process, permitting real-time adjustments in their utilization, particularly when confronted with errors.
> 2. An alternative category of research (e.g., ReAct [2]) has formulated distinct formal actions tailored to particular tasks, signifying the actions undertaken by LLMs, including reasoning and tool manipulation. However, these techniques tend to excessively prioritize tool manipulation and lack the capacity for sustained reasoning akin to CoT. By contrast, our approach seamlessly aligns with the conversational flow of multi-turn dialogues, enabling uninterrupted CoT reasoning while also facilitating timely tool manipulation when required.
>
> For the concern about the baseline method in the experiment, we do not report the performance of finetuned smaller language model, as previous work [3] has shown that small language model cannot achieve better performance than LLMs, especially for very complex mathematical reasoning tasks (e.g., MATH). Besides, for other large-model-based methods, we select two recently proposed popular approaches LP [4] and PHP [5], and PHP is the previous SOTA in MATH benchmark. As the results shown in our paper, our approach outperforms all these baseline methods.
>
> To provide a comprehensive view about the effectiveness of our approach, we report the performance of our approach and three LLM-based methods, and also the finetuned language model LLaMA-13B. We present the results in the following table:
>
> \\begin{array}{|l|c|c|c|}
>         \\hline
>         \\textbf{Method} & \\textbf{Algebra} & \\textbf{PA} & \\textbf{NT}  \\\\
>         \\hline
>         \\textrm{ChatGPT + ChatCoT (Ours)} & \\textbf{56.1} & \\textbf{59.2}  & \\textbf{32.6} \\\\
>         \\hline
>         \\textrm{ChatGPT + ComplexCoT [6]} & 51.6 & 53.7 & 30.0 \\\\
>         \\textrm{ChatGPT + PAL} & 34.3 & 50.5 & 31.3 \\\\
>         \\textrm{ChatGPT + ReAct} & 54.5 & 54.5 & 31.7 \\\\
>         \\hline
>         \\textrm{LLaMA-13b + Fintune} & 5.7 & 7.0 & 4.1 \\\\
>         \\hline
> \\end{array}
>
> Reference
>
> [1] PAL: Program-aided Language Models
>
> [2] ReAct: Synergizing Reasoning and Acting in Language Models
>
> [3] Measuring Mathematical Problem Solving With the MATH Dataset
>
> [4] Learning to Program with Natural Language
>
> [5] Progressive-Hint Prompting Improves Reasoning in Large Language Models
>
> [6] Complexity-based prompting for multi-step reasoning

---

### Official Review · Reviewer_GffQ · 2023-08-03

**Soundness:** 3

**Excitement:**

4: Strong: This paper deepens the understanding of some phenomenon or lowers the barriers to an existing research direction.

**Paper Topic And Main Contributions:**

The paper presents ChatCoT, a tool-augmented chain-of-thought reasoning framework designed to enhance the reasoning abilities of large language models (LLMs) for complex tasks that involve specific knowledge and multi-hop reasoning. The authors model the chain-of-thought reasoning as multi-turn conversations, allowing LLMs to interact with external tools or perform reasoning at each turn. The paper is well-written, presenting the ChatCoT framework and its components in a clear and concise manner. The step-by-step explanation of the iterative tool-augmented reasoning process enhances the paper's readability.

**Questions For The Authors:**

1. In lines 243-246 of the paper, it is mentioned that "Note that when the input expression or equation is ill-formed or unsolved, the above tools would return an error." Does the detection of ill-formed input solely rely on ChatGPT? If there are errors in the CoTs generated by ChatGPT, how can such errors be mitigated?

2. In lines 378-379 of the paper, it is stated, "To solve this sub-problem, which tool can we use?" Is the selection of the appropriate tool entirely dependent on ChatGPT? Similarly, in the case of potential misjudgments by ChatGPT, how can such decisions be adjusted or corrected?

**Reasons To Accept:**

The introduction of ChatCoT as a tool-augmented chain-of-thought reasoning framework is a novel contribution to the field. By combining multi-turn conversations and external tools, ChatCoT leverages the strengths of chat-based LLMs in a unified manner, effectively tackling complex reasoning tasks that require multi-hop reasoning and specific knowledge.

**Reasons To Reject:**

The paper does not address how the system handles situations where ChatGPT generates incorrect reasoning chains. It is essential to explore the robustness of the framework and whether it has built-in mechanisms to recognize errors in the generated chains and adjust the reasoning direction accordingly. This raises questions about the framework's fault tolerance and the ability of ChatGPT to self-correct when faced with erroneous reasoning paths.

**Reproducibility:**

3: Could reproduce the results with some difficulty. The settings of parameters are underspecified or subjectively determined; the training/evaluation data are not widely available.

**Reviewer Confidence:**

4: Quite sure. I tried to check the important points carefully. It's unlikely, though conceivable, that I missed something that should affect my ratings.

---

> ### Author Rebuttal · Authors · 2023-08-28
>
> We sincerely thank the reviewer for the insightful suggestion and appreciate the positive feedback.
>
> $\textbf{Response-to-Reason2Reject}$
>
> In this work, we are seeking a more unified way to integrate CoT reasoning and tool manipulation, to perform a continuous reasonable reasoning process with frequent tool manipulation. In our approach, we do not explicitly design special strategies to handle situations where ChatGPT generates incorrect reasoning chains but propose several ways to improve our approach for preventing the above issue to some extent. We list them as follows:
> 1. We design the conversational memory mechanism that provides task knowledge, tools knowledge, and reasoning format knowledge through the conversation history for LLMs. In this way, LLMs can find useful knowledge in the context to prevent making mistakes.
> 2. We designed the iterative tool-augmented reasoning procedure to decompose the one-pass reasoning into several reasoning steps. In each step, LLMs can freely choose to perform natural language reasoning or call tools, where tools can help accurately perform arithmetic operations that LLMs are not good at.
>
> Although we have devised the above strategies to improve the performance and our ablation study in Table 4 has shown their effectiveness, it is still hard to perfectly steer LLMs to not generate any errors. Recently, there have been a number of works that explore the way to identify and revise the content generated by LLMs, such as self-consistency [1] and self-refine [2]. In fact, these methods are vertical to our work and can be integrated with ours to further improve our performance. As shown in the following Table, we implement self-consistency (SC) and self-refine (SR) on our approach and report the relative performance improvement in the bracket. The experimental results have shown that they can both improve our performance on complex reasoning tasks.
>
> \\begin{array}{|l|c|}
>         \\hline
>         \\textbf{Method} & \\textbf{CP} & \\textbf{NT}  \\\\
>         \\hline
>         \\textrm{ChatCoT + SC} & 40.1_{+5.9\\%} & 38.3_{+5.7\\%} \\\\
>         \\textrm{ChatCoT + SR} & 36.9_{+2.7\\%} & 33.2_{+0.6\\%} \\\\
>         \\hline
> \\end{array}
>
> Based on the results, we also conduct the error analysis on randomly sampled several problems, to show which type of errors has been corrected by the SR approach.
>
> \\begin{array}{|c|c|c|c|}
>         \\hline
>         \\textbf{Logic Error} & \\textbf{Ill-Formed Equation
> } & \\textbf{Middle Steps Missing
> } & \\textbf{Misunderstanding of Problem
> }  \\\\
>         \\hline
>         40\\% & 25\\% & 20\\% & 15\\% \\\\
>         \\hline
> \\end{array}
>
> As shown in the above table, we can observe that LLMs can self-correct a variety of the mistakes they made in the reasoning process, and address more logic errors. It shows that it is promising to recognize errors in the reasoning chains via LLMs. We leave the further investigation into our future work.
>
> To better show how LLMs correct the mistakes through ChatCoT, we present an example in the following and the reasoning procedure generated from LLMs which is italic.
>
> > [Tool Knowledge]
> >
> > [Retrieval-Augmented Task Knowledge]
> >
> > [Multi-turn Reasoning Format]
> >
> > $\\textbf{User}$: Problem: What is the remainder when $13^{13} +5$ is divided by 6? Let's think step by step
> >
> > $\\textbf{LLM}$: *First, we need to find out the value of $13^{13}+5$ using the properties of exponents.*
> >
> > ......
> >
> > $\\textbf{LLM}$: *Second, we need to find out the remainder when $302875106592258$ is divided by $6$.*
> >
> > $\\textbf{User}$: To solve this sub-problem, which tool can we use?
> >
> > $\\textbf{LLM}$: *Calculator*
> >
> > $\\textbf{User}$: Give me the equation to calculate
> >
> > $\\textbf{LLM}$: *$302875106592258 \\mod 6$*
> >
> > $\\textbf{User}$: Results: $302875106592258 \\mod 6 = 1.81725063955355e+15*mod$ Continue reasoning
> >
> > $\\textbf{LLM}$: *Finally, we know that the remainder when $13^{13}+5$ is divided by 6 is the same as the remainder when $302875106592258$ is divided by 6, which is $\\boxed{2}$*
> >
> > $\\textbf{User}$: The solution above might some mistakes, you should check the solution and get the final answer.
> >
> > $\\textbf{LLM}$: *Apologies for the mistake. Let's correct the solution. To find the remainder when $13^{13} + 5$ is divided by 6, we can simplify the expression $13^{13}$ first. Using the property of remainders, we can observe that $13 \\equiv 1 \\pmod{6}$. Therefore, $13^{13} \\equiv 1^{13} \\equiv 1 \\pmod{6}$. Now, we can rewrite the expression as $13^{13} + 5 \\equiv 1 + 5 \\equiv 6 \\equiv \\boxed{0} \\pmod{6}$. Hence, the remainder when $13^{13} + 5$ is divided by 6 is 0.*
>
> In this example, we can observe that LLMs try to calculate the exact value of $13^{13}+5$ at first, then get the remainder. But calculator does not under the expression $302875106592258 \\mod 6$ and returns the wrong error values. Through refining, LLM changes the solution and gets the correct answer.
>
> $\\textbf{Response-to-Question1}$
>
> Here, we only focus on the ill-formed input of the interface, which can be detected by the abnormal returns of the interfaces, with no need for using ChatGPT. Indeed, we can utilize LLMs like ChatGPT to check the generated CoT or input expressions. However, it will increase the cost of crafting prompts via human annotators and the computation cost of LLMs. Thus, we only consider utilizing the abnormal returns to detect the ill-formed input.
>
> Besides, as we discuss above, there are a number of works that explores the way to refine the generated reasoning process by LLMs, such as self-consistency [1] and self-refine [2]. These methods can be easily integrated with our approach, and our experimental results in the above table have shown that the combination of our approach and these methods can lead to further performance improvement.
>
> $\\textbf{Response-to-Question2}$
>
> The selection of proper tools is a complex decision-making problem that requires the LLM to well plan the reasoning process and foresee the expected results after manipulating the tool. Therefore, our approach incorporates ChatGPT to fulfill this subtask.
>
> In our approach, once LLMs misselect an improper tool, the abnormal return would remind the LLM to correct the mistake and make a new decision. Besides, our designed iterative Step-by-Step Reasoning might also help correct the mistakes in intermediate steps. If there are any errors in the last reasoning step, LLMs would have an opportunity to discover them and try to revise them with newly selected tools. All the above strategies help improve the reasoning performance of our approach.
>
> Reference:
>
> [1] Self-Consistency Improves Chain of Thought Reasoning in Language Models
>
> [2] Self-Refine: Iterative Refinement with Self-Feedback

---

### Official Review · Reviewer_X74Y · 2023-08-04

**Soundness:** 3

**Excitement:**

2: Mediocre: This paper makes marginal contributions (vs non-contemporaneous work), so I would rather not see it in the conference.

**Paper Topic And Main Contributions:**

This  manuscript proposes to use tools to enhance the reasoning capabilities of LLMs by Chain-of-Thought. The experiment results on two complex reasoning datasets MATH and HotpotQA achieve better results.


**Reasons To Accept:**

The method proposed by  this manuscript may be helpful in improving the reasoning ability of LLMs.


**Reasons To Reject:**

Although the author considers tools manipulation by LLMs as the interaction between LLMs and tools, this doesn't seem to be different from calling tools frequently. In the perspective of the author's understanding, LLMs calling tools are regarded as part of the reasoning process. If the tool needs to be invoked frequently, the method proposed by the author still requires frequent access to the tool, which is not fundamentally different from previous tool enhancements.


**Reproducibility:**

3: Could reproduce the results with some difficulty. The settings of parameters are underspecified or subjectively determined; the training/evaluation data are not widely available.

**Reviewer Confidence:**

4: Quite sure. I tried to check the important points carefully. It's unlikely, though conceivable, that I missed something that should affect my ratings.

---

> ### Author Rebuttal · Authors · 2023-08-28
>
> We sincerely thank the reviewer for the insightful suggestion.
>
> In our work, we aim to integrate the chain-of-thought (CoT) prompting strategy with frequent tool manipulation operations for improving the reasoning ability of LLMs but not interrupting the reasoning process of CoT. To achieve this goal, the most essential challenge is to guarantee the consistency of the CoT reasoning process after calling tools. However, the one-pass generation manner of the vanilla CoT method does not support it. Thus, we consider revising it as the multi-turn conversation manner, where each utterance is one intermediate step of the whole reasoning process and the invocation of tools can be freely inserted after the current utterance when necessary.
>
> Besides, in our approach, we also devise other novel strategies to further improve its performance. First, as the whole reasoning process follows the way of multi-turn conversation, we propose the conversational memory mechanism that provides the useful knowledge for LLMs in the early turns of the conversation, including the tool knowledge, retrieval-augmented task knowledge, and the multi-turn reasoning format. Second, we design proper prompts for guiding LLMs to leverage the knowledge within the conversational memory, to iteratively perform reasoning with tools, until reaching the answer.
>
> To clearly show the difference between our approach and existing methods, we also roughly categorize existing work about tool enhancement into the following two types, where we introduce them briefly and individually present their divergences with ours:
> 1. Several work has utilized the planning-then-execution paradigm to enhance the reasoning ability of LLMs. In this kind of work （e.g., PAL [1]）, LLMs are required to generate the plan of the whole reasoning procedure, where tools should be arranged in proper steps for manipulation. Then, the planned procedure would be executed to complete the reasoning process, where LLMs cannot interact with tools or change the plan after generating it, even seeing obvious mistakes. As a comparison, in our approach, LLMs can freely interact with tools in the whole reasoning process and adjust its utilization when meeting errors.
> 2. Another kind of work (such as ReAct [2]) has designed formal actions pertaining to specific tasks, which indicate the operations of LLMs, e.g., reasoning and tool manipulation. These methods over-emphasize tool manipulation and can not perform continuous reasoning like CoT. As a comparison, our approach can follow the dialog flow of the multi-turn conversation, to perform continuous CoT reasoning and meanwhile manipulate tools when necessary.
> Besides, our past experiments have verified that ChatCoT is better than other tool enhancement methods. We report the results in the following table:
>
> \\begin{array}{|l|c|c|c|}
>         \\hline
>         \\textbf{Method} & \\textbf{Algebra} & \\textbf{PA} & \\textbf{NT}  \\\\
>         \\hline
>         \\textrm{ChatGPT + ChatCoT (Ours)} & \\textbf{56.1} & \\textbf{59.2}  & \\textbf{32.6} \\\\
>         \\textrm{ChatGPT + CoT w/ Tool} & 35.9 & 40.5 & 19.4 \\\\
>         \\textrm{ChatGPT + PAL} & 34.3 & 50.5 & 31.3 \\\\
>         \\textrm{ChatGPT + ReAct} & 54.5 & 54.5 & 31.7 \\\\
>         \\hline
> \\end{array}
>
> As shown in the table, our approach outperforms CoT w/ Tool, PAL and ReAct, the representative work of the above two categories.
>
> Reference:
>
> [1] PAL: Program-aided Language Models
>
> [2] ReAct: Synergizing Reasoning and Acting in Language Models

---

### Meta-Review · Area_Chair_RVLb · 2023-09-27

**Recommendation:** 3

**Metareview:**

Overall, the reviewers find the proposed approach valuable for improving complex reasoning tasks but raise concerns about its novelty, robustness, and the need for broader comparisons in the experimental evaluation. The authors have provided a fair amount of discussion to the reviewers' concerns in the rebuttal phase.

---

### Decision · Program_Chairs · 2023-10-07

**Decision:**

Accept-Findings

**Comment:**

Overall, the reviewers find the proposed approach valuable for improving complex reasoning tasks but raise concerns about its novelty, robustness, and the need for broader comparisons in the experimental evaluation. The authors have provided a fair amount of discussion to the reviewers' concerns in the rebuttal phase.